# Pain in the brain: Psychological correlates of chronic pain and fibromyalgia

**Marcello Passarelli**[1], **Luca Rizzi**[2], **Laura Casetta**[2], **Vittorio Guerrieri**[3]*, **Diego Rocco**[4], **Raffaella Perrella**[5]

1 Institute of Educational Technology, National Research Council, Genoa, Italy, 2 Associazione Centro di Psicologia e Psicoterapia Funzionale, Padua, Italy, 3 Department of Psychology, University of Milano-Bicocca, Milan, Italy, 4 Department of Developmental and Social Psychology, University of Padua, Padua, Italy, 5 Department of Psychology, University of Campania "Luigi Vanvitelli", Caserta, Italy

* v.guerrieri4@campus.unimib.it

**Data availability statement:** All data and R codes are available from the GitHub database (https://github.com/M-Pass/Pain_in_the_brain).

**Funding:** The author(s) received no specific funding for this work.

## Abstract

Fibromyalgia is a chronic disorder causing widespread pain, fatigue, and cognitive issues, linked to central nervous system dysfunction. This cross-sectional study compared the psychological profiles of women with fibromyalgia (N = 76) and women with non-fibromyalgic chronic pain (N = 73). Using cluster analysis, scale score comparisons using MANCOVA, and correlation difference tests, we examined whether these conditions present distinct psychological profiles in terms of touch avoidance, anxiety, depression, psychotic symptoms, self-criticism, pain acceptance, pain impact, and positive emotions. Cluster analysis successfully differentiated between FM and non-FM CP patients with 76% accuracy with and an adjusted Rand index of.26, suggesting distinct psychological profiles while acknowledging substantial overlap. FM patients reported significantly higher levels of anxiety, depression, and psychotic symptoms, along with greater self-criticism and reduced activity engagement ($p < .001$ for all significant comparisons expect psychotic symptoms, $p = .07$, and opposite sex touch avoidance, $p = .014$). Both groups showed elevated touch avoidance, with no significant differences between them. Correlation analysis revealed that self-criticism played a more central role in FM patients' psychological distress, showing stronger associations with depression, anxiety, and positive emotions compared to non-FM CP patients (Jennrich test for differences between the two correlation matrices: $X^2 = 436.72$, $p < .001$). These findings suggest that while FM and non-FM CP share common psychological features, FM patients experience more severe psychological distress, with self-criticism potentially playing a crucial maintaining role. Results suggest that psychological interventions for FM patients should prioritize reducing self-criticism and enhancing self-reassurance, while healthcare providers should be mindful of the significant psychological burden these patients carry.

**Competing interests:** The authors have declared that no competing interests exist.

## Introduction

Chronic pain (CP) is a highly prevalent condition, affecting approximately 20% of the population in both the US and Europe [1,2], with 1 in 20 people newly diagnosed each year [3]. More recent estimates suggest that the prevalence may be as high as 28% in Italy [4] and up to 40% in Europe, where it has been described as "a European epidemic" due to its substantial impact on quality of life and associated psychological distress [5].

CP is defined in the ICD-11 classification as pain that persists or recurs for more than three months [6], typically progressing from an acute to a chronic condition, when pain loses its usefulness as an indicator of injury.

Importantly, the ICD-11 classification for CP considers localization and aetiology but also introduces a category for primary CP, described as "pain in one or more anatomical regions that persists or recurs for longer than three months and is associated with significant emotional distress or significant functional disability […] that cannot be better explained by another chronic pain condition" [6]. One of the main manifestations of primary CP is fibromyalgia (FM), a condition characterized by widespread musculoskeletal pain with no clear aetiology and spontaneous fluctuations [7]. FM has a particularly severe psychological impact, even when compared to other CP conditions, and its diagnostic criteria include fatigue and sleep disturbances. FM is estimated to affect 2.7% of the European and 3.6% of the Italian population [8]. Both CP and FM disproportionately affect women [4]. The gender gap is especially prominent in FM, with a meta-analysis reporting a prevalence of 3.98% in women and 0.01% in men [8], although the refinement of American College of Rheumatology diagnostic criteria over time seems to have highlighted a more balanced, yet still highly skewed, distribution [9,10].

The ICD-11 definition of CP puts clear emphasis on the physical sensation of pain, which is the main symptom typically reported by these patients. However, CP has an important psychological component that should not be underestimated, and which is even more evident in FM patients. Pain is a significant stressor that can greatly increase an individual's psychological burden [11]. When pain is severe, it can negatively impact patients' quality of life, limiting daily activities and exacerbating pre-existent mood disorders such as depression and anxiety. Indeed, CP increases the risk for depression up to 4 times [12,13], and those suffering from major depressive disorder are three times more likely to experience non-neuropathic pain and six times more likely to report neuropathic pain [14]. On average, 52% of CP patients meet the criteria for depression [15], and these individuals are two to three times more likely to report suicidal behaviours [16].

The combination of depression or anxiety and CP is associated with increased pain intensity, greater disability [17], and higher health care usage and costs compared to having either condition alone [18].

The high prevalence of depression among CP patients can be explained by the Integrated Chronic Pain Model (ICPM; [19]). According to the ICPM, physical and social pain share overlapping characteristics, and chronic physical pain alone can

lead to psychosocial consequences. In this context, social pain is defined as a specific emotional response to the perception of being excluded from desired relationships or devalued by desired relationship partners or groups [20–22].

According to the ICPM, physical pain can result in self-criticism [23], loss of confidence and self-esteem [19], and feelings of exclusion or ostracism [24]. From an evolutionary perspective, the social component of physical pain might serve an adaptive function by prompting help-seeking behaviour [24]. When help-seeking does not result in symptom relief or a sense of social connection and acceptance, this can lead to feelings of hopelessness, further contributing to depression and anxiety, and ultimately affecting overall well-being, daily functioning, and quality of life [25].

The aim of this study is to explore and compare the psychological profiles of individuals with FM and those with non-fibromyalgic CP. The following sections will outline the key psychological factors associated with CP and FM based on current literature, along with areas where we anticipate differences.

## Psychosocial aspects of CP and FM

As FM is a subtype of CP, its psychological manifestations largely mirror those of other CP conditions. However, some notable differences—particularly in the severity of psychological symptoms—emerge when comparing FM patients with the general CP population. For instance, depression is more common among FM patients, with a prevalence of 63% [26] compared to 52% in the broader CP population. Similarly, generalised anxiety disorder occurs in 9.1% of FM patients [26] versus 4.4–6.9% in those with non-FM CP [27]. Additionally, FM patients generally report higher levels of pain and disability, more somatic complaints, lower well-being, less acceptance of pain [28], and an overall poorer quality of life [29].

In addition, some studies have identified elevated levels of self-criticism in women with FM, possibly related to body image [30]. FM patients also tend to exhibit more rumination and somatisation, even when employing the same pain coping strategies as non-FM CP patients. Furthermore, they experience poorer outcomes following Cognitive-Behavioural Therapy interventions [31], while compassion-focused approaches seem to be effective [32,33]. Finally, FM patients often report anhedonia towards social touch [34], whereas evidence for this phenomenon in non-FM CP patients is mixed [35]. For these reasons, we anticipate self-criticism to play a more important role in patients with FM than in patients with CP, and social touch to be less impactful for these patients.

## Rationale of the study

In many healthcare settings within Western societies, pain management continues to predominantly follow a biomedical approach, focusing on the organic causes of pain while undervaluing its psychological and psychosocial dimensions [36]. At the same time, studies have highlighted a tendency among healthcare professionals to attribute pain complaints to psychological distress [37]. This often leads to an underestimation of the severity of patients' pain, with clinicians sometimes interpreting psychological distress as the primary cause of pain rather than acknowledging it as part of the physical pain experience [19]. This dismissal of CP patients' experiences is critical, as feelings of being ignored, humiliated, or devalued are strongly correlated with the development of depression, anxiety, anger, and self-criticism [38].

In this regard, FM is an especially interesting case study, as it is a condition without objective, observable diagnostic markers. The diagnosis of FM can take years, with patients often consulting numerous medical specialists before receiving a definitive diagnosis [39]. The subjective nature of FM symptoms and the absence of clear biological markers have hampered its understanding, its treatment within healthcare, and its social acceptance. FM patients frequently report feeling dismissed by healthcare professionals, often to a greater extent than patients with other forms of CP [40], and delayed diagnosis is associated with more severe symptoms and overall disease severity [41].

Given this, the experience of CP in patients whose pain can be swiftly diagnosed through biological markers may differ—socially and psychologically—from that of FM patients, who often face delayed diagnosis and greater scepticism regarding their symptoms.

In order to improve healthcare interventions for these populations, it is important to raise awareness about the psychological aspects of physical pain, particularly how these may manifest differently in FM patients compared to other CP patients. We would expect patients with FM to not only exhibit more signs of distress, but also stronger associations between the physical and the social components of pain, as a result of the social stigma they typically experienced.

When comparing the psychological profiles of individuals with FM and those with non-FM CP, we will focus on domains where CP is known to have an impact: depressive symptoms, anxiety, self-criticism, and affective quality. Additionally, we will consider touch avoidance—which we expect to be more pronounced in FM patients—along with pain acceptance and its impact on daily life. Lastly, we will explore whether the two profiles differ in psychiatric symptoms other than anxiety and depression.

Our investigation is guided by the following research question: "How do the FM and CP populations differ in terms of psychological profiles?". Thus, we formulated three hypotheses to test:

• H1: The two populations form two natural clusters when considering their psychological characteristics.

• H2: There are significant differences between the two groups with respect to psychological variables.

• H3: The correlation patterns between the psychological variables of the two populations differ.

Given the difficulties in recruiting men with FM, we limited our investigation to women for both FM and non-FM CP.

## Method

### Participants

The study involved a total of 149 women: 76 with FM (age range: 23–78 years, mean = 50.22 ± 10.47 years) and 73 with non-FM, localized CP (age range: 23–88 years, mean = 59.82 ± 16.24 years). Participants were recruited through convenience sampling in Italian hospitals and clinics from the 26th of November 2023 to the 16th of April 2024. FM and CP diagnosis, classified according to the 2016 ACR criteria [42], was provided by medical staff employed in the involved clinics (specifically rheumatologists, for FM patients). Due to a statistically significant age difference between the two groups (p < .001), all comparisons were adjusted for age (see Data Analysis section) and missing data (1.28%) was imputed using multivariate imputation by chained equations [43]. Usage of *localized* CP as the comparison group ensures that no undiagnosed FM patients are inadvertently included in this group. The study was approved by the ethical committee of the University of Padua (Ref. # 4044, 06/03/21). All participants were informed about the study procedures and signed an informed consent form. The study was carried out in compliance with the competent committee on human experimentation's ethical criteria, as well as the Helsinki Declaration of 1975, as updated in 1983. No financial support was received for this study.

### Measures and procedure

Participants were informed about the study's purpose and, upon providing informed consent, completed a battery of questionnaires administered via LimeSurvey. The order of the questionnaires was randomised for each participant. For the non-FM CP group, references to "fibromyalgia" in the Fibromyalgia Impact Questionnaire were substituted with "chronic pain". The battery included the following instruments:

### Touch Avoidance Questionnaire (TAQ)

The TAQ [44,45] is a 37-items questionnaire which assesses the level of aversion towards physical touch, using a response scale ranging from 1 (I completely disagree) to 5 (I completely agree). The questionnaire has 5 subscales measuring touch aversion towards different categories of individuals: romantic partners (10 items), same-sex or opposite-sex friends (6 items each), family members (6 items), and strangers (3 items). Six items are unscored. A sample item is "I

often find it unbearable to be touched by my partner." Higher scores denote higher touch avoidance. The internal reliability is good for the Partner, Same Sex, Opposite Sex, and Family subscales (Cronbach's α ranging.84−.91 in the validation study,.83−.89 in our sample). Reliability is lower, and potentially critical, for the 3-item Stranger subscale (Cronbach's α = .59 in the Italian validation study,.41 in our sample).

### State-Trait Anxiety Inventory Y-2 (STAI Y-2)

The STAI Y-2 [46,47] is a 20-item questionnaire that assesses trait anxiety using a Likert scale ranging from 1 ("almost never") to 4 ("almost always"), with higher scores indicating higher levels of anxiety. The questionnaire has no subscales. Sample items include "I feel inadequate" and "I wish I could be as happy as others seem to be". Internal reliability for the STAI Y-2 is good, with Cronbach's α reported as.85−.90 in the Italian validation study and.90 in our sample.

### Beck Depression Inventory – II (BDI-II)

The BDI-II [48,49] is a 21-item self-report questionnaire assessing the severity of depression symptoms. Each item is rated on a scale from 0 to 3, with higher scores indicating more severe depressive symptoms. The questionnaire does not include subscales. A sample item ("pessimism") uses the scale "I am not discouraged about my future" (score 0), "I feel more discouraged about my future than I used to" (score 1), "I do not expect things to work out for me" (score 2), and "I feel my future is hopeless and will only get worse" (score 3). Internal reliability for the BDI-II is good, with Cronbach's α = .92 in the Italian validation study and.90 in our sample.

### 16-item Prodromal Questionnaire (iPQ-16)

The iPQ-16 [50,51] is a 16-item screening tool used to identify individuals at high risk of developing a psychotic disorder, with strong psychometric properties for screening help-seeking populations in general mental health care settings [52]. This measure was included to investigate whether FM patients differ from CP patients in the prevalence of psychiatric symptoms beyond those typically associated with anxiety or depression. Each item (e.g., "I have sometimes been undecided whether some things I have experienced were real or imaginary") can be marked as "true" or "false". The full score is the sum of all items marked as true. Higher scores indicate higher risk of developing a psychotic disorder. Internal reliability for the iPQ-16 is good, with Cronbach's α = .81 in the Italian validation study and.81 in our sample.

### Forms of Self-Criticizing/Attacking and Self-Reassuring Scale (FSCRS)

The FSCRS [53,54] is a 22-item questionnaire that assesses self-criticism and the ability to reassure oneself during difficult times. Participants respond using a Likert scale ranging from 0 ("Not at all true for me") to 4 ("Totally true for me"). The measure includes three subscales: Hated-Self (5 items, such as "I feel a sense of disgust with myself"), Inadequate-Self (9 items such as "I feel disappointed with myself easily"), and Reassured-Self (8 items such as "I can remind myself of my qualities"). Internal reliability for the FSCRS is good in the Italian validation study, with Cronbach's α = .84 (Hated-Self),.86 (Inadequate-Self),.90 (Reassured-Self). Reliability is lower for the Hated-Self subscale in our sample, with Cronbach's α = .70,.88,.89 for the three subscales, respectively.

### Chronic Pain Acceptance Questionnaire (CPAQ)

The CPAQ [55,56] is a 20-item questionnaire designed to measure pain acceptance, using a response scale ranging from 0 ("Never true") to 6 ("Always true"). It comprises two subscales: Activity Engagement (11 items), that is, the will to engage in activities despite pain, and Pain Willingness (9 items), i.e., accepting pain instead of trying to avoid or control it. Higher scores indicate a greater ability to engage in activities despite pain and to accept pain. Internal reliability for the CPAQ is

acceptable, with Cronbach's α of.83 (Activity Engagement) and.76 (Pain Willingness) in the Italian validation study and.87 and.78, respectively, in our sample.

### Fibromyalgia Impact Questionnaire (FIQ)

The Fibromyalgia Impact Questionnaire (FIQ) [8,57] assesses health status in individuals with FM, focusing on physical health, psychological distress, pain, sleep, fatigue, and overall well-being. It includes 10 items that evaluate a patient's ability to perform daily tasks, such as cooking and walking, with responses ranging from 0 ("Always able to do") to 3 ("Never able to do"), and the average of these responses forms the physical functioning subscale score. Two additional items ask patients to indicate the number of days in the past week they felt good and missed work due to their condition. Seven other items assess the ability to perform one's job, pain, fatigue, morning tiredness, stiffness, anxiety, and depression, using a 0–10 scale, with 10 indicating maximum impairment. Scores for physical functioning, "days felt good," and "days missed work" are rescaled to a 0–10 range. The FIQ total score is calculated by summing the rescaled physical functioning score, number of days *not* felt good, number of workdays missed, and the scores of the seven additional items, resulting in a total score ranging from 0 to 100, with 100 indicating the maximum negative impact on daily life. Internal reliability for the FIQ is very high, with Cronbach's α of.94 for the physical functioning subscale, and.90 for the total questionnaire in the Italian validation study. In our study, Cronbach's α = .84 (physical functioning) and.89 (full questionnaire).

### Dispositional Positive Emotion Scales (DPES)

The DPES [58,59] is a self-report tool measuring individuals' predispositions to experience positive emotional states. The questionnaire comprises 37 items, with responses given on a Likert scale from 1 ("strongly disagree") to 7 ("strongly agree"). It includes six subscales: Happiness (11 items) represents a positive attitude towards life in general (e.g., "I am an intensely cheerful person"); Compassion (5 items) reflects the desire to care for others, especially those who are vulnerable or needy (e.g., "Taking care of others gives me a warm feeling inside"); Amusement (5 items) relates to the ability to experience humor (e.g., "The people around me make a lot of jokes"); Love (6 items) addresses how individuals conceive of closeness and authenticity (e.g., "I love many people"); Pride (5 items) pertains to the social image of oneself (e.g., "Many people respect me"); and Awe (5 items) involves the ability to connect with something greater (e.g., "I see beauty all around me"). In the original version, there is a seventh scale called "Contentment" with 5 items, which is incorporated into the "Happiness" scale in the Italian form. Internal reliability for the DPES is acceptable in the Italian validation study, with Cronbach's α ranging.75−.91. In our sample, we note substantially lower αs, with the lowest being.66 (Love), followed by.71 (Awe),.72 (Compassion),.78 (Pride),.79 (Amusement) and.90 (Happiness).

### Data analysis

Our analysis of the psychological profiles of patients with FM and non-FM CP will involve three steps:

1. To address H1 we will perform a cluster analysis, a technique that identifies natural groupings within the data without prior labelling. This analysis will help determine whether the FM and non-FM CP groups can be distinguished based on their psychological scale scores alone. If distinct clusters emerge, this could suggest that the psychological profiles of these groups are inherently different, providing further evidence for unique psychological characteristics associated with FM. We will perform this analysis using ensemble clustering, a robust and reliable – if compute-intensive – technique that effectively combines different clustering approaches in a data-driven way [60,61]. We will examine the extent to which a two-cluster solution matches the division between participants with FM and those with non-FM CP. To account for the age difference between the two groups, clustering will use only age-corrected subscale scores as input; the correction will be applied by regressing each subscale on age and using the regression residuals as input for the

ensemble clustering, thereby removing the confounding effect of age [62]. Clustering will be performed using the diceR package [63] following the recommendations of Gao et al. [64]. Specifically, we will perform 250 resamplings using 80% of the data and performing clustering using k-means, partition around medoids, hierarchical clustering with Ward linkage, hierarchical density-based spatial clustering of applications with noise, Gaussian mixture model, non-negative matrix factorization, and spectral clustering (the selection of algorithms will aim to maximise diversity, in line with the ensemble clustering approach). We will employ Euclidean distance, linkage clustering ensemble as the consensus function, and reweigh the algorithms so that those with the most consistent results across resamplings will be favoured by the consensus function. All these choices are guided by the principle of maximising clustering stability.

2. Second, to address H2 we will compute and compare the (sub)scale scores for the two groups across all questionnaires. The comparison will be carried out via a multivariate analysis of covariance (MANCOVA) using all subscale scores as outcomes, the disorder as a predictor, and age as a covariate. This approach will allow us to determine whether and where the two groups differ, while accounting for the significant age difference between them as well as the correlation between outcome measures. Alongside MANCOVA results and estimated effect size, we will also report the overlapping index η [65] for both the raw data and the age-corrected data. This is a distribution-independent index that quantifies the shared area between two probability density functions (from 0 to 100%) and will be, alongside clustering results, an indication of how much the two populations are similar.

3. Finally, to test H3, we will investigate whether the pattern of correlations between variables differs between the two groups by testing for their difference and observing the absolute difference between the two correlation matrices. For instance, we may find similar scores between the groups in terms of pain acceptance and depressive symptoms, yet observe that these variables are correlated in one group but not in the other. To compare the correlational pattern, after computing the pairwise correlation matrices and partializing for age, we will use the Jennrich test [66,67], which evaluates their difference using the chi square statistic.

The reporting of this study conforms to STROBE guidelines [68] Data and R codes are available at https://github.com/M-Pass/Pain_in_the_brain.

## Results

### Descriptive statistics

Descriptive statistics for questionnaire scores are reported in Table 1. A graphical summary of subscale scores is shown in Fig 1. Note that in Fig 1, to aid readability, all scores have been rescaled to have a theoretical range of 0–1.

### Clustering

The best clustering results were achieved using k-means, non-negative matrix factorization, and Gaussian mixture model clustering (see Fig 2 for a consensus matrix example; the rest of consensus matrices are reported in the Supporting information).

The resulting solution partially aligns with participants' disorder: cluster 1 includes 16 participants with non-FM CP and 57 with FM, while cluster 2 includes 57 patients with non-FM CP and 20 with FM. This corresponds to an overall accuracy of 76% and an adjusted Rand index of.26. While this result suggests that there is partial overlapping in the psychological profile of the two populations, the clustering correctly identify diagnosis with a probability well above chance. This shows that the two populations are, to some extent, distinct and distinguishable.

### Comparisons on scale scores

Results of the MANCOVA are presented in Table 2.

**Table 1. Descriptive statistics for all subscale scores, divided for participants with FM and participants with non-FM CP (SD = Standard deviation).**

| Scale | Group | Mean ± SD | Median | Range | Skewness | Kurtosis |
|---|---|---|---|---|---|---|
| TAQ Partner | FM | 3.56 ± .71 | 3.50 | 1.80 - 5.00 | .09 | −.51 |
| | non-FM CP | 3.78 ± .84 | 3.80 | 1.00 - 5.00 | −.88 | .64 |
| TAQ Family | FM | 3.01 ± .97 | 3.00 | 1.17 - 5.00 | .11 | −.92 |
| | non-FM CP | 3.08 ± 1.13 | 3.17 | 1.00 - 5.00 | −.21 | −.84 |
| TAQ Same Sex | FM | 4.02 ± .70 | 4.08 | 2.17 - 5.00 | −.31 | −.73 |
| | non-FM CP | 3.99 ± .99 | 4.20 | 1.33 - 5.00 | −.83 | −.04 |
| TAQ Opposite Sex | FM | 3.62 ± .85 | 3.50 | 1.00 - 5.00 | −.39 | −.13 |
| | non-FM CP | 3.26 ± 1.13 | 3.17 | 1.00 - 5.00 | −.28 | −.73 |
| TAQ Stranger | FM | 3.17 ± .87 | 3.00 | 1.00 - 5.00 | −.19 | −.42 |
| | non-FM CP | 3.21 ± .83 | 3.33 | 1.00 - 5.00 | −.08 | −.19 |
| STAI Y-2 | FM | 56.49 ± 10.80 | 58.50 | 32.00 - 78.00 | −.30 | −.44 |
| | non-FM CP | 44.48 ± 10.19 | 42.00 | 23.00 - 74.00 | .84 | .21 |
| BDI | FM | 19.38 ± 7.41 | 18.00 | 6.00 - 37.00 | .47 | −.28 |
| | non-FM CP | 10.18 ± 4.88 | 10.00 | 0.00 - 20.00 | −.09 | −.63 |
| IPQ | FM | 7.32 ± 3.56 | 7.00 | 0.00 - 14.00 | −.01 | −.85 |
| | non-FM CP | 5.44 ± 3.65 | 5.00 | 0.00 - 14.00 | .39 | −.64 |
| FSCRS Hated-Self | FM | 1.83 ± .72 | 1.60 | 1.00 - 4.40 | 1.41 | 1.94 |
| | non-FM CP | 1.42 ± .47 | 1.20 | 1.00 - 2.60 | .98 | .01 |
| FSCRS Inadequate-Self | FM | 2.85 ± .76 | 2.78 | 1.11 - 4.78 | .29 | −.24 |
| | non-FM CP | 2.06 ± .79 | 2.00 | 1.00 - 3.78 | .42 | −.86 |
| FSCRS Reassured-Self | FM | 3.03 ± .70 | 2.94 | 1.38 - 4.75 | .22 | −.12 |
| | non-FM CP | 3.87 ± .91 | 4.12 | 1.25 - 5.00 | −.73 | −.27 |
| CPAQ PW | FM | 19.11 ± 8.75 | 18.00 | 2.00 - 45.00 | .38 | .00 |
| | non-FM CP | 21.01 ± 11.43 | 21.00 | 0.00 - 43.00 | −.14 | −1.05 |
| CPAQ AE | FM | 30.22 ± 11.40 | 31.00 | 6.00 - 51.00 | −.21 | −.72 |
| | non-FM CP | 43,23 ± 12.38 | 46.00 | 6.00 - 62.00 | −.86 | .35 |
| FIQ Physical Functioning | FM | 4.17 ± 2.14 | 4.33 | 0.00 - 8.67 | −.10 | −.60 |
| | non-FM CP | 38.09 ± 14.33 | 2.00 | 0.00 - 9.33 | 1.76 | 4.31 |
| FIQ Total Score | FM | 66.55 ± 12.86 | 66.67 | 21.86 - 90.9 | −.39 | .39 |
| | non-FM CP | 38.09 ± 14.33 | 36.10 | 13.43 - 72.24 | .38 | −.68 |
| DPES Happiness | FM | 3.58 ± 1.17 | 3.50 | 1.55 - 6.00 | .26 | −1.16 |
| | non-FM CP | 4.91 ± 1.09 | 5.00 | 1.91 - 6.91 | −.44 | −.18 |
| DPES Pride | FM | 4.54 ± 1.24 | 4.70 | 1.60 - 7.00 | −.39 | −.54 |
| | non-FM CP | 5.72 ± .98 | 5.80 | 2.80 - 7.00 | −.81 | .30 |
| DPES Love | FM | 4.14 ± 1.17 | 4.17 | 1.50 - 6.17 | −.33 | −.78 |
| | non-FM CP | 4.21 ± 1.03 | 4.33 | 2.17 - 6.50 | .05 | −.75 |
| DPES Compassion | FM | 6.23 ± .69 | 6.20 | 4.40 - 7.00 | −.74 | −.20 |
| | non-FM CP | 5.27 ± .64 | 6.40 | 4.40 - 7.00 | −.98 | .27 |
| DPES Amusement | FM | 4.42 ± 1.40 | 4.40 | 1.00 - 6.80 | −.16 | −.96 |
| | non-FM CP | 4.26 ± 1.40 | 4.40 | 1.00 - 7.00 | −.41 | −.38 |
| DPES Awe | FM | 4.51 ± 1.30 | 4.60 | 1.00 - 6.80 | −.42 | −.46 |
| | non-FM CP | 4.43 ± 1.21 | 4.40 | 1.20 - 7.00 | −.32 | −.28 |

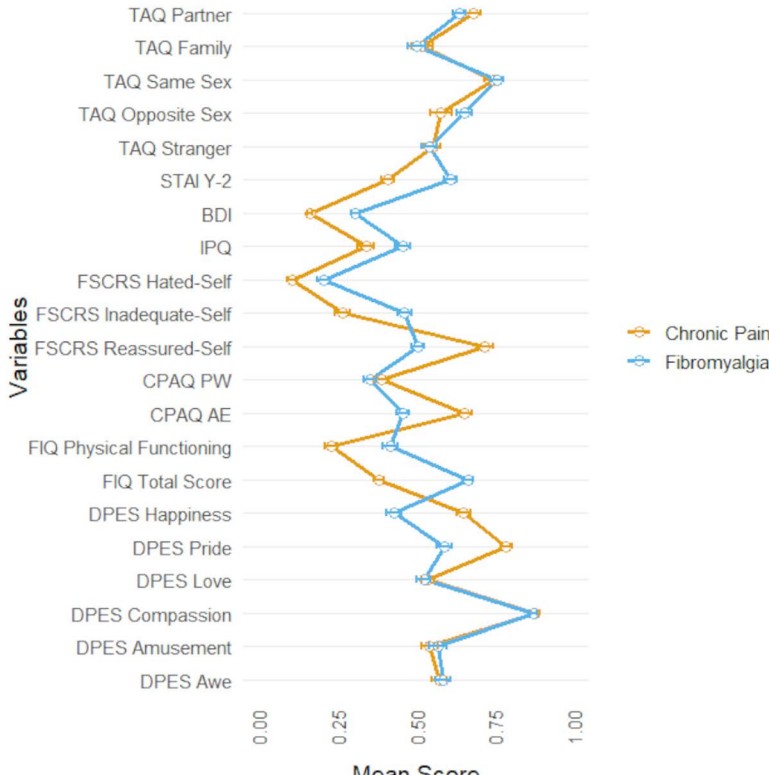

**Fig 1. Comparison of scale scores between the two groups.** To aid readability, all scale scores have been rescaled to have a theoretical range of 0-1. The left side of the graph shows group means with 95% confidence intervals, while the right side of the graph shows boxplots for each variable in the two groups.

On average, participants with FM reported higher levels of touch avoidance towards opposite-sex friends, as well as increased anxiety, depressive symptoms, symptoms prodromal to psychotic disorders, a sense of self-hatred and inadequacy, and greater impact of pain on daily life (FIQ). Concurrently, these participants demonstrated a lower tendency to self-reassure, decreased activity engagement, and less frequent experiences of happiness and pride. No statistically significant differences were observed between the two groups across all subscales of touch avoidance (except for opposite-sex friends), willingness to accept pain, and the frequency of experiencing certain positive emotions, such as love, compassion, amusement, and awe. The distribution overlap % was still relatively high even for the most significant results (e.g., 42% for the FIQ), indicating that while population-level means are different, many participants present scores that would be compatible with both a FM and non-FM CP profile.

## Comparisons on scale correlations

Finally, we compared the two correlation matrices to assess the differences in correlation patters. The Jennrich test highlighted significant differences between the two matrices ($X^2 = 436.72$, $p < .001$) showing that the two groups not only present different psychological profiles, but that the various traits interact differently with one another.

A side-by-side comparison of the correlation matrices for FM and non-FM CP patients is presented in Fig 3. While the two matrices appear largely similar, several notable differences between certain correlations are evident. For instance, there is a strong negative correlation between the self-criticism subscales of kindness and inadequacy in FM patients

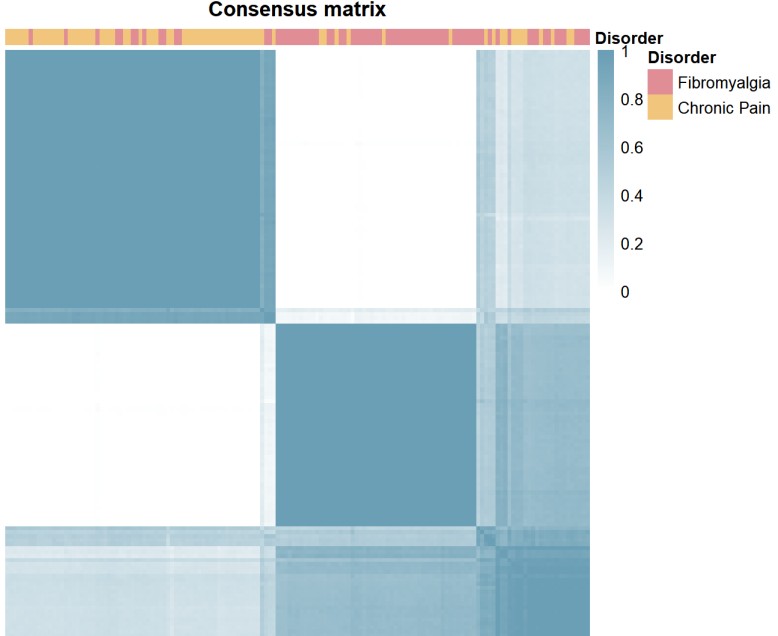

**Fig 2. Consensus matrix for one of the clustering algorithms considered, showing a high degree of clustering stability.**

($r = -.68$). In contrast, this relationship is not present in CP patients ($r = -.14$). Generally, many correlations exist among FM patients that are not replicated in the non-FM CP sample. These differences will be further examined in the Discussion.

## Discussion

This study explored the primary psychological characteristics of patients with FM and non-FM CP, with a focus on identifying similarities and differences between the typical profiles of these two patient populations. The investigation was conducted by assessing whether an unsupervised machine learning algorithm would cluster participants according to their diagnosis, comparing scale scores, and examining correlations between scales.

The cluster analysis results underscore the meaningful psychological differences between FM and non-FM CP patients. The psychological profiles of these two groups could be differentiated prior to knowing their diagnosis, with the clustering algorithm correctly classifying 76% of patients. This finding provides evidence that FM and non-FM CP patients present distinct psychological profiles, which suggests that these groups should receive differentiated support. On the other hand, the fact that 24% of the patients were classified in the 'wrong' cluster should not be understated; results suggest that the *typical* profile of these patient populations is different, but there is enough within-group variability that a patient with FM could be similar to the typical non-FM CP patient, and vice versa.

In terms of scale scores, significant differences were observed, particularly in the severity of physical and psychological symptoms. Patients with FM generally reported feeling worse and experiencing greater impairment, as indicated by the Fibromyalgia Impact Questionnaire (FIQ) score, which measures impairment in daily life rather than pain intensity. FM patients presenting relatively higher levels of anxiety, depressive symptoms, and symptoms prodromal to psychotic disorders align with previous literature, which consistently reports greater psychological distress among these patients compared to those with non-FM CP [26]. Interestingly, patients with FM also demonstrated lower levels of self-reassurance and a reduced tendency to engage in activities despite their pain, which may further exacerbate their psychosocial distress.

**Table 2. MANCOVA results for subtest score differences according to disorder (FM/ non-FM CP) using age as a covariate. \*=p<.05; \*\*=p<.01; \*\*\*=p<.001 (df=degrees of freedom).**

| Scale | t (df=146) | p-value | Partial η² | Overlap %/ Age-corrected overlap % |
|---|---|---|---|---|
| TAQ Partner | −1.61 | .109 | .01 (negligible) | 78%/ 82% |
| TAQ Family | −.59 | .555 | .00 (negligible) | 92%/ 90% |
| TAQ Same Sex | .54 | .588 | .00 (negligible) | 87%/ 86% |
| **TAQ Opposite Sex** | **2.49** | **.014 \*** | **.04 (small)** | **82%/ 82%** |
| TAQ Stranger | −.01 | .996 | .00 (negligible) | 92%/ 92% |
| **STAI Y-2** | **5.65** | **<.001 \*\*\*** | **.26 (large)** | **54%/ 61%** |
| **BDI** | **7.92** | **<.001 \*\*\*** | **.35 (large)** | **49%/ 58%** |
| **IPQ** | **2.71** | **.007 \*\*** | **.06 (medium)** | **82%/ 86%** |
| **FSCRS Hated-Self** | **3.57** | **<.001 \*\*\*** | **.10 (medium)** | **79%/ 79%** |
| **FSCRS Inadequate-Self** | **4.87** | **<.001 \*\*\*** | **.22 (large)** | **68%/ 72%** |
| **FSCRS Reassured-Self** | **−5.22** | **<.001 \*\*\*** | **.23 (large)** | **60%/ 67%** |
| CPAQ PW | −1.54 | .126 | .00 (negligible) | 80%/ 80% |
| **CPAQ AE** | **−6.67** | **<.001 \*\*\*** | **.23 (large)** | **58%/ 59%** |
| **FIQ Physical Functioning** | **5.51** | **<.001 \*\*\*** | **.19 (large)** | **52%/ 56%** |
| **FIQ Total Score** | **11.74** | **<.001 \*\*\*** | **.53 (large)** | **35%/ 42%** |
| **DPES Happiness** | **−6.28** | **<.001 \*\*\*** | **.26 (large)** | **59%/ 65%** |
| **DPES Pride** | **−5.50** | **<.001 \*\*\*** | **.22 (large)** | **65%/ 69%** |
| DPES Love | −.34 | .736 | .01 (negligible) | 92%/ 91% |
| DPES Compassion | .12 | .908 | .00 (negligible) | 92%/ 90% |
| DPES Amusement | −.16 | .875 | .00 (negligible) | 89%/ 90% |
| DPES Awe | .51 | .614 | .00 (negligible) | 92%/ 91% |

Lastly, we observe no difference between the two groups in touch avoidance, except for a minor difference in the Opposite Sex scale; however, it is noteworthy that both groups exhibited exceptionally high levels of touch avoidance compared to the validation sample [69]. While this result is somewhat expected in FM patients, who experience difficulties to discriminate between gentle and fast touch [70], it is more surprising in non-FM CP patients, as we would expect them to use social touch as a source of comfort.

According to the Integrated Chronic Pain Model [19], the social component of physical pain might serve an adaptive function by prompting help-seeking behavior [24]. Our data suggest that, in some ways, social touch is not a source of comfort. There are two possible explanations for this phenomenon: it may not lead to symptom relief, or it may expose the person to a sense of shame correlated with a low level of self-acceptance. Shame is closely associated with self-criticism and affects the expression of symptoms, the ability to disclose painful information, various forms of avoidance (e.g., dissociation and denial), and difficulties in help-seeking [71]. Shame is typically linked to the experience of having negative aspects of oneself exposed [72,73]. In the case of FM and non-FM CM, the body may be the object of self-attacking because it is perceived as broken and socially undesirable, which may be central to the internal shame response and lead to a tendency toward touch avoidance.

Finally, the analysis of differences in correlations between scales provides further insights. The pattern of correlations between the two groups is largely similar; however, we observe some important differences, typically involving all sub-scales of the FSCRS, the Pride subscale of the DPES, and/or the Stranger subscale of the TAQ. Specifically, FM patients showed stronger interconnections between self-criticism, depressive symptoms, anxiety, and certain positive emotions (such as pride, love, and happiness). This suggests that self-criticism—and particularly a lack of self-kindness—plays a critical role in maintaining and possibly exacerbating psychological symptoms. This finding aligns with the scale score comparisons for the DPES, where FM patients reported feeling less pride and happiness, while the other four positive

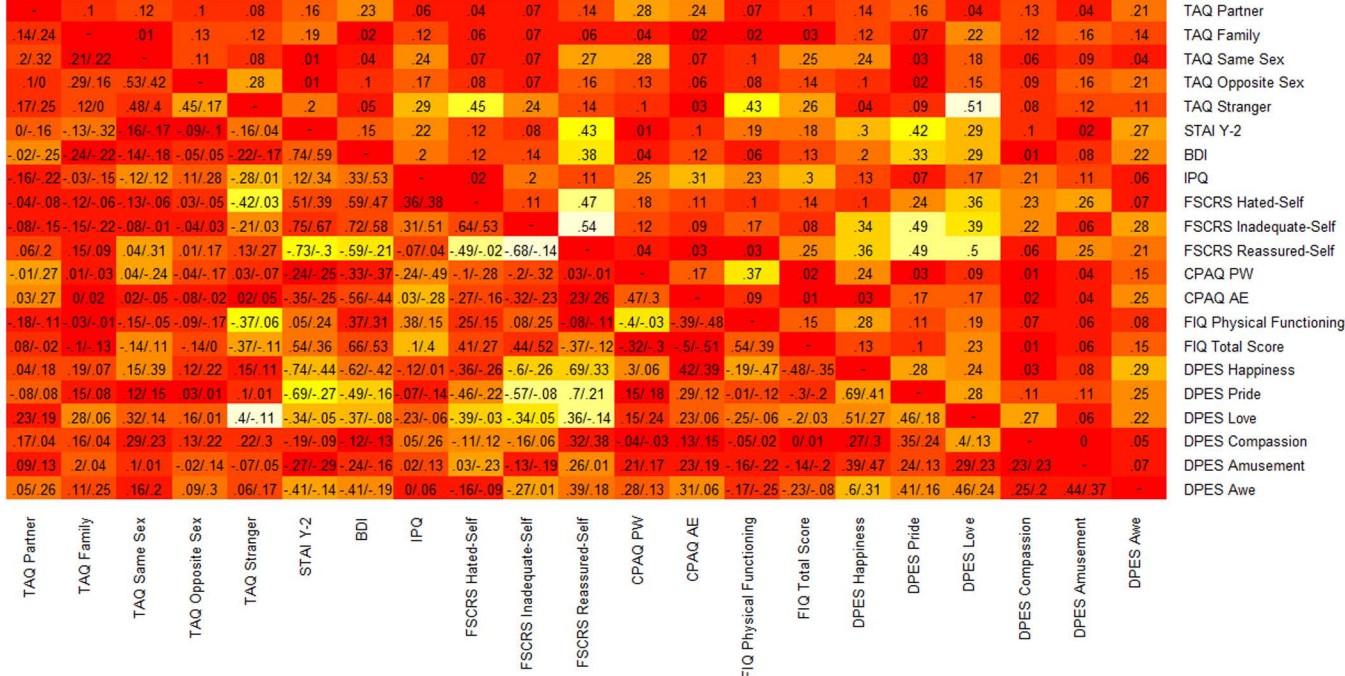

**Fig 3. Heatmap of the comparison between the two correlation matrices.** The colour represents the absolute difference between the values. The lighter the colour, the bigger the difference. In the lower triangle, cells report the correlations for patients with FM and non-FM CP, in that order. In the upper triangle, cells report the absolute difference between correlations observed in the two samples.

emotions (love, compassion, amusement, and awe) were no less frequent than in non-FM CP patients. Happiness and pride are closely linked to self-evaluation and satisfaction with achieving personal goals, while the other positive emotions are associated with a sense of connection to others or the world. Joy is also related to gratitude, which involves recognizing support or benefits received from others. Therefore, heightened self-criticism tends to diminish feelings of joy and pride but does not similarly impact the other positive emotions. In the FM population, the DPES happiness scale and the FSCRS inadequacy scale show a negative correlation, possibly indicating difficulty in maintaining a positive outlook on life and the future due to low self-esteem and uncertainty about one's abilities.

A potential explanation for these results may lie in the diagnostic process of FM. As discussed in the introduction, FM patients often experience longer delays before receiving a diagnosis compared to non-FM CP patients, as FM is primarily a diagnosis of exclusion [74–76]. During this process, FM patients frequently report feeling dismissed, disbelieved, and humiliated by healthcare professionals and their social environment [8,19,31]. There is increasing evidence that self-evaluation is closely tied to expectations of how others will evaluate us—in other words, the feeling that others might view us with contempt or ridicule can significantly impact our own self-perception [77]. It is possible that FM patients internalise the social stigma associated with their condition, leading to heightened self-criticism and a stronger association between self-criticism and both physical and psychological distress. This hypothesis is further supported by the correlations between the DPES Pride subscale and the FSCRS subscales: these correlations were high in FM patients but lower in non-FM CP patients. As the DPES Pride subscale includes items related to social recognition (e.g., "many people respect me" and "people usually recognise my authority"), the strong link between pride and self-criticism in FM patients may provide indirect evidence that their self-criticism is influenced by perceived social stigma. In this context, therapeutic approaches focused on reducing self-criticism and enhancing self-compassion appear particularly promising for supporting FM patients [23,30].

One of these approaches is Compassion-Focused Therapy (CFT), which belongs to third-wave therapies, alongside Acceptance and Commitment Therapy (ACT). These therapies emphasize awareness and acceptance of emotional distress rather than focusing on mastery or control of symptoms [78]. The CFT model was developed by Paul Gilbert and it aims to cultivate compassion as an alternative to the automatic responses of self-criticism and shame in response to distress. To achieve this, it employs mindfulness exercises, guided imagery, and experiential exercises like therapeutic writing. Recent reviews [78,79] indicate that CFT is a promising intervention, particularly effective for individuals who are highly self-critical, as the model acknowledges the substantial impact of shame and self-criticism on emotional regulation. While CFT has been applied in both healthcare and group settings, there is still limited literature regarding its effects on CP management, specifically in the context of FM. Gooding et al. [80] suggest that compassion-based interventions can help patients with pain to better regulate their emotions, thereby improving pain management. Consistent with this, studies have found that in individuals with persistent pain, self-compassion predicts the extent of pain-related disability and how effectively they cope with it [81,82]. Furthermore, Penlington [83] reports that a compassion-focused intervention for chronic pain can reduce pain-related distress, pain intensity, anxiety, and depression, while increasing self-efficacy. Based on the results of this study, we would expect CFT to be even more effective for patients with FM.

Finally, it is worth noting that symptoms related to psychosis (as measured by the IPQ) were less correlated with daily impairment and attitudes towards pain in FM patients than in non-FM CP patients. This suggests that the higher levels of psychosis-related symptoms in FM patients do not stem from experiencing more intense pain or greater impairment. This contrasts with the non-FM CP group, who, despite reporting similar levels of physical pain, present a lower IPQ score and appear to retain a greater ability to engage in daily activities.

## Conclusions

This study examined psychological differences between FM and non-FM CP patients, highlighting greater distress in the former. The study has some important limitations: a relatively small sample size, low internal reliability of certain tests, such as the Touch Avoidance Questionnaire and the Dispositional Positive Emotions Scale, and being limited to women. Still, the fact it identified relatively distinct psychological profiles of CP and FM patients despite these limitations suggests that the difference between the psychological manifestation of these two conditions is present and clinically relevant.

Future research should explore whether this heightened self-criticism stems from internalised stigma associated with the disorder, and whether social pain is causally linked to psychological symptoms in FM patients. Acknowledging and addressing the significant psychological and social pain these patients experience may be essential in psychological interventions aimed at improving their quality of life. Such interventions should prioritise fostering self-reassurance, reducing self-criticism, and enhancing patients' ability to engage in daily activities despite their pain. Moreover, healthcare professionals should be educated on the importance of validating FM patients' experiences to alleviate the harmful effects of social pain. Adequate pain treatment is a human right, and healthcare systems must provide it—not only by addressing physical pain, but by recognising its social and psychological.

## Supporting information

**S1 Data. Consensus matrices**
(ZIP)

## Author contributions

**Conceptualization:** Marcello Passarelli, Luca Rizzi, Laura Casetta, Diego Rocco, Raffaella Perrella.

**Data curation:** Marcello Passarelli, Luca Rizzi, Laura Casetta, Diego Rocco.

**Formal analysis:** Marcello Passarelli, Vittorio Guerrieri.

**Funding acquisition:** Raffaella Perrella.

**Investigation:** Luca Rizzi, Laura Casetta, Diego Rocco.

**Methodology:** Marcello Passarelli, Vittorio Guerrieri.

**Supervision:** Raffaella Perrella.

**Visualization:** Marcello Passarelli, Vittorio Guerrieri.

**Writing – original draft:** Marcello Passarelli, Luca Rizzi, Laura Casetta, Vittorio Guerrieri.

**Writing – review & editing:** Marcello Passarelli, Luca Rizzi, Laura Casetta, Vittorio Guerrieri.

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
