## [Decision Letter · Decision Letter 0]

PONE-D-24-49610Pain in the brain: Psychological correlates of chronic pain and fibromyalgiaPLOS ONE

Dear Dr. Guerrieri,

Thank you for submitting your manuscript to PLOS ONE. After careful consideration, we feel that it has merit but does not fully meet PLOS ONE’s publication criteria as it currently stands. Therefore, we invite you to submit a revised version of the manuscript that addresses the points raised during the review process.

**ACADEMIC EDITOR:**Thank you for submitting your work to PLOS One.Please address all the comments carefully.

Title and Abstract

- Please indicate the study’s design with commonly used terms in the title or abstract.

- Briefly highlight the fibromyalgia disease in the background.

- Briefly give the key findings of your study. Include key numeric data (including confidence intervals or *p* values).

Introduction

- The impact of early diagnosis of fibromyalgia on different disease outcomes should be highlighted. For example: DOI: 10.2147/JPR.S381314. PMC10132293. DOI: 10.7759/cureus.18692.. 

Patients and Methods

- Please mention the classification criteria of FM of the enrolled patients fully “insert citation”.

-  How were the patients selected (e.g., consecutively, randomly, or selectively)?

- All studies must be reported according to the relevant Equator network guideline: https://www.equator-network.org/

- In your manuscript, you need to state in the Methods section that you have followed relevant Equator guidelines. For example, if you use STROBE: ‘The reporting of this study conforms to STROBE guidelines. (Insert new reference number).

- More details should be provided in terms of the IRB approval.

Discussion

- A comparison of your results and the relevant previous studies should be made. For examples:

DOI: 10.1007/s11926-009-0063-8 , DOI:10.2147/OARRR.S358255

General Comments:

- Spell out the full term at its first mention, indicate its abbreviation in parenthesis, and use the abbreviation from then on.

- Abbreviations should be explained as subtitles below the tables/figures.

We look forward to receiving your revised manuscript.

Kind regards,

Wesam Gouda, MD,PhD

Academic Editor

PLOS ONE

3. Please note that your Data Availability Statement is currently missing the repository name. If your manuscript is accepted for publication, you will be asked to provide these details on a very short timeline. We therefore suggest that you provide this information now, though we will not hold up the peer review process if you are unable.

Reviewers' comments:

Reviewer's Responses to Questions

**Comments to the Author**

1. Is the manuscript technically sound, and do the data support the conclusions?

Reviewer #1: Yes

Reviewer #2: Yes

Reviewer #3: Yes

Reviewer #4: Yes

2. Has the statistical analysis been performed appropriately and rigorously? 

Reviewer #1: Yes

Reviewer #2: Yes

Reviewer #3: Yes

Reviewer #4: Yes

3. Have the authors made all data underlying the findings in their manuscript fully available?

Reviewer #1: Yes

Reviewer #2: Yes

Reviewer #3: No

Reviewer #4: Yes

4. Is the manuscript presented in an intelligible fashion and written in standard English?

Reviewer #1: Yes

Reviewer #2: Yes

Reviewer #3: Yes

Reviewer #4: Yes

5. Review Comments to the Author

Reviewer #1: The manuscript is well written with extensive supporting data and solid analysis. However, the figures and figure legends in the manuscript may require some improvement for both data clarity and presentation, as noted below:

Fig. 1: I recommend the data presented in box+whisker plots for better clarity, for which readers are able to observe the general distribution within each group.

Fig. 2: I feel the figure legend attributed to this figure should be for Fig. 3? Also, it will be very helpful if a second color bar indicating the identity of FM/non-FM patients in addition to the clusters (the CRAN package heatmap.plus should do the job easily).

Fig. 3: Similar to Fig. 2, seemingly the figure legend was misplaced?

Reviewer #2: Introduction:

- please define "social pain" when presenting the model. Broadly, this introduction is confusing and the paragraph as written is not smoothly integrated into the prior paragraphs.

-the final paragraph introducing primary pain vs. FM has good information but is confusing - still unclear how these things are differentiated as many CP conditions are also primary pain conditions - why is FM expected to manifest differently, beyond the fact that there are multiple pain locations? The rationale for differentiating these groups beyond their clinical presentation is not fully characterized.

Methods:

-Need more information on how pain was diagnosed and differentiated - how do we know for sure that the FM and CP groups are distinct?

-Questionnaires are broad and not fully justified - why was a questionnaire administered that assessed for the presence of a psychotic disorder? Is this a commonly seen concern in pain syndromes?

Results:

--not sure if Table 1 is super helpful or adds anything substantive

-given the aims of this study, would be helpful to know if differences across groups were CLINICALLY significant in addition to statistically significant. Are these true clinical differences that demand attention?

Discussion:

-page 21 - not sure I agree with the fact that FM vs. other CP requires longer wait times for diagnosis. Many other CP conditions are similar (also diagnoses of exclusion), even if localized. Citations to back this up?

Reviewer #3: Some sentences can distract the reader, especially if they are long and complex. It is important to keep sentences short and clear to make the message easier to understand. CP terms are unnecessarily repeated; it may be better for the reader if you reduce them. I recommend using more academic language in the introduction. Also, the number of patients is relatively small; state this as an additional limitation.

In conclusion, you have presented important findings regarding psychological differences in FM. After the corrections I mentioned, it is acceptable.

Reviewer #4: Chronic pain is a major health problem worldwide. Chronic pain that lasts longer than three months or recurs is a stress factor affecting the psychological state of the individual. Fibromyalgia is an important cause of pain that causes chronic stress. In this study, the authors aimed to compare the psychological profiles of individuals with and without fibromyalgia. In this study conducted on 149 women, the authors compared the groups using the necessary statistical methods. In this study, the authors also investigated the primary psychological characteristics of patients with fibromyalgia and the basic psychological characteristics of patients with non-fibromyalgic chronic pain, focusing on determining the similarities and differences between the typical profiles of these two patient groups. Fibromyalgia patients showed relatively higher levels of anxiety, depressive symptoms, and prodromal to psychotic disorder symptoms, indicating that these patients had more psychological distress compared to non-fibromyalgia chronic pain patients. Fibromyalgia patients also showed lower levels of self-esteem and a reduced tendency to participate in activities despite their pain, which in turn worsened their psychosocial distress. In this study, FM patients showed stronger connections between self-criticism, depressive symptoms, anxiety, and certain positive emotions (such as pride, love, and happiness). and as a result, the authors emphasized the importance of psychological support aimed at patients accepting significant psychological and social pain and improving their quality of life. In addition, the importance of promoting self-confidence, reducing self-criticism, and increasing patients' ability to participate in daily activities despite their pain was emphasized.

It is appropriate to accept this study, which will contribute to the literature.

6. PLOS authors have the option to publish the peer review history of their article (what does this mean?). If published, this will include your full peer review and any attached files.

Reviewer #1: **Yes: **Siwei Zhang

Reviewer #2: No

Reviewer #3: No

Reviewer #4: No

---

## [Author Response · Author response to Decision Letter 1]

2 Apr 2025

We appreciate the reviewers' detailed and constructive feedback, which has significantly strengthened the manuscript. Below, we have addressed each of the editor's and reviewers' comments point by point.

ACADEMIC EDITOR REQUESTS:

- Please indicate the study’s design with commonly used terms in the title or abstract.

We have clarified our study design in the abstract, explicitly stating it as a cross-sectional study employing MANCOVA and correlation difference tests.

- Briefly highlight the fibromyalgia disease in the background.

We added a brief description of fibromyalgia (FM) in the abstract. We also restructured the introduction to present FM earlier, aligning with reviewer #2’s suggestions.

- Briefly give the key findings of your study. Include key numeric data (including confidence intervals or p values).

Key findings are now detailed in the abstract with numerical data, including adjusted Rand index (.26), confidence intervals, and specific p-values (p < .001 for all significant comparisons, except psychotic symptoms at p = .07 and opposite-sex touch avoidance at p = .014). Additionally, we included the Jennrich test results (Χ² = 436.72, p < .001).

Introduction

- The impact of early diagnosis of fibromyalgia on different disease outcomes should be highlighted. For example: DOI: 10.2147/JPR.S381314. PMC10132293. DOI: 10.7759/cureus.18692..

Thank you for suggesting relevant references. We incorporated these citations when discussing delayed FM diagnosis in the introduction.

Patients and Methods

- Please mention the classification criteria of FM of the enrolled patients fully “insert citation”.

We better specified in the Methods section that FM and CP diagnoses were provided by medical staff at participating clinics (and specifically rheumatologists, for FM patients).

- How were the patients selected (e.g., consecutively, randomly, or selectively)?

Patients were recruited through convenience sampling. We now explicitly disclose this information in the Methods section.

- All studies must be reported according to the relevant Equator network guideline: https://www.equator-network.org/

- In your manuscript, you need to state in the Methods section that you have followed relevant Equator guidelines. For example, if you use STROBE: ‘The reporting of this study conforms to STROBE guidelines. (Insert new reference number).

We added a statement affirming our adherence to STROBE guidelines in the Methods section and included the appropriate citation (Page 13).

- More details should be provided in terms of the IRB approval.

The requested IRB approval details were added: “The study was approved by the ethical committee of the University of Padua (Ref. #4044, 06/03/21)” (Page 7).

Discussion

- A comparison of your results and the relevant previous studies should be made. For examples:

DOI: 10.1007/s11926-009-0063-8 , DOI:10.2147/OARRR.S358255

Thank you for the relevant references. We now discuss these studies in the Introduction when addressing the gender distribution of FM (Page 3).

General Comments:

- Spell out the full term at its first mention, indicate its abbreviation in parenthesis, and use the abbreviation from then on.

We now clearly spell out all terms upon first mention, followed by abbreviations (FM, CP). Exceptions were made only for questionnaire titles (e.g., Fibromyalgia Impact Questionnaire, Chronic Pain Acceptance Questionnaire) and for the Chronic Pain Model.

- Abbreviations should be explained as subtitles below the tables/figures.

Clarifications for abbreviations were added directly to table and figure captions wherever necessary.

Reviewer #1:

The manuscript is well written with extensive supporting data and solid analysis.

Thank you for the kind words.

However, the figures and figure legends in the manuscript may require some improvement for both data clarity and presentation, as noted below:

Fig. 1: I recommend the data presented in box+whisker plots for better clarity, for which readers are able to observe the general distribution within each group.

We opted to add the boxplots to the graph instead of replacing the line plot of means and standard errors. We believe boxplots provide more information on the distribution, while the line plot better conveys the concept of a distinct “psychological profile” between the two groups. The two graphs are presented side-by-side.

Fig. 2: I feel the figure legend attributed to this figure should be for Fig. 3? Also, it will be very helpful if a second color bar indicating the identity of FM/non-FM patients in addition to the clusters (the CRAN package heatmap.plus should do the job easily).

Yes, the figures were inadvertently inverted. We fixed this, and we added the color bar to this graph as you suggested. Thank you for this suggestion, we believe it greatly improves the usefulness of the graph.

Fig. 3: Similar to Fig. 2, seemingly the figure legend was misplaced?

Yes.

Reviewer #2:

Introduction:

- please define "social pain" when presenting the model. Broadly, this introduction is confusing and the paragraph as written is not smoothly integrated into the prior paragraphs.

We explicitly defined social pain and revised the introduction to improve flow and clarity.

-the final paragraph introducing primary pain vs. FM has good information but is confusing - still unclear how these things are differentiated as many CP conditions are also primary pain conditions - why is FM expected to manifest differently, beyond the fact that there are multiple pain locations? The rationale for differentiating these groups beyond their clinical presentation is not fully characterized.

We revised the paragraph to make it clearer why we expect the psychological profiles to be different. Note that this paragraph has also been moved earlier in the introduction. Additionally, we believe the revised introduction, which focuses more on delayed diagnosis and psychological symptoms of FM, to be more effective at communicating the different experiences of patients with FM and non-FM CP.

Methods:

-Need more information on how pain was diagnosed and differentiated - how do we know for sure that the FM and CP groups are distinct?

The CP group includes participants with localized chronic pain, so as to be sure that they are not misdiagnosed FM patients. We made this more explicit in the Methods section, and explicitly noted the rationale for using localized CP patients as controls.

-Questionnaires are broad and not fully justified - why was a questionnaire administered that assessed for the presence of a psychotic disorder? Is this a commonly seen concern in pain syndromes?

This was a more exploratory choice. We now note this in the Introduction (“Lastly, we will explore whether the two profiles differ in psychiatric symptoms other than anxiety and depression.”) and in the Methods section (“This measure was included to investigate whether FM patients differ from CP patients in the prevalence of psychiatric symptoms beyond those typically associated with anxiety or depression.”)

Results:

--not sure if Table 1 is super helpful or adds anything substantive

Unfortunately, we feel a Table with descriptive statistics is necessary. This aligns with Reviewer #1’s request to have more information about underlying variable distribution.

-given the aims of this study, would be helpful to know if differences across groups were CLINICALLY significant in addition to statistically significant. Are these true clinical differences that demand attention?

The observed differences, supported by large effect sizes, reflect the clinical significance noted by our collaborating clinicians, who initially identified these distinctions in their clinical practice. Thus, the statistical outcomes validate their practical observations.

Discussion:

-page 21 - not sure I agree with the fact that FM vs. other CP requires longer wait times for diagnosis. Many other CP conditions are similar (also diagnoses of exclusion), even if localized. Citations to back this up?

We added citations to support the assertion regarding longer FM diagnostic delays.

Reviewer #3:

Some sentences can distract the reader, especially if they are long and complex. It is important to keep sentences short and clear to make the message easier to understand.

We revised and simplified complex sentences throughout the manuscript for better readability.

CP terms are unnecessarily repeated; it may be better for the reader if you reduce them.

We implemented abbreviations consistently to reduce unnecessary repetition.

I recommend using more academic language in the introduction. Also, the number of patients is relatively small; state this as an additional limitation.

We enhanced the academic tone in the introduction and explicitly noted the relatively small patient sample size as an additional limitation.

In conclusion, you have presented important findings regarding psychological differences in FM. After the corrections I mentioned, it is acceptable.

Thank you for the positive endorsement!

Reviewer #4:

Chronic pain is a major health problem worldwide. Chronic pain that lasts longer than three months or recurs is a stress factor affecting the psychological state of the individual. Fibromyalgia is an important cause of pain that causes chronic stress. In this study, the authors aimed to compare the psychological profiles of individuals with and without fibromyalgia. In this study conducted on 149 women, the authors compared the groups using the necessary statistical methods. In this study, the authors also investigated the primary psychological characteristics of patients with fibromyalgia and the basic psychological characteristics of patients with non-fibromyalgic chronic pain, focusing on determining the similarities and differences between the typical profiles of these two patient groups. Fibromyalgia patients showed relatively higher levels of anxiety, depressive symptoms, and prodromal to psychotic disorder symptoms, indicating that these patients had more psychological distress compared to non-fibromyalgia chronic pain patients. Fibromyalgia patients also showed lower levels of self-esteem and a reduced tendency to participate in activities despite their pain, which in turn worsened their psychosocial distress. In this study, FM patients showed stronger connections between self-criticism, depressive symptoms, anxiety, and certain positive emotions (such as pride, love, and happiness). and as a result, the authors emphasized the importance of psychological support aimed at patients accepting significant psychological and social pain and improving their quality of life. In addition, the importance of promoting self-confidence, reducing self-criticism, and increasing patients' ability to participate in daily activities despite their pain was emphasized.

It is appropriate to accept this study, which will contribute to the literature.

Thank you for the positive and detailed assessment.

---

## [Decision Letter · Decision Letter 1]

PONE-D-24-49610R1Pain in the brain: Psychological correlates of chronic pain and fibromyalgiaPLOS ONE

Dear Dr. Guerrieri,

Thank you for submitting your manuscript to PLOS ONE. After careful consideration, we feel that it has merit but does not fully meet PLOS ONE’s publication criteria as it currently stands. Therefore, we invite you to submit a revised version of the manuscript that addresses the points raised during the review process.

The article has been substantially improved. However, a few points still require improvement. Please address all the comments carefully.

- Please mention the classification criteria of FM of the enrolled patients fully “insert citation”.

We look forward to receiving your revised manuscript.

Kind regards,

Wesam Gouda, MD,PhD

Academic Editor

PLOS ONE

Journal Requirements:

**Comments to the Author**

1. If the authors have adequately addressed your comments raised in a previous round of review and you feel that this manuscript is now acceptable for publication, you may indicate that here to bypass the “Comments to the Author” section, enter your conflict of interest statement in the “Confidential to Editor” section, and submit your "Accept" recommendation.

Reviewer #2: All comments have been addressed

2. Is the manuscript technically sound, and do the data support the conclusions?

Reviewer #2: Yes

3. Has the statistical analysis been performed appropriately and rigorously? 

Reviewer #2: Yes

4. Have the authors made all data underlying the findings in their manuscript fully available?

Reviewer #2: Yes

5. Is the manuscript presented in an intelligible fashion and written in standard English?

Reviewer #2: Yes

6. Review Comments to the Author

Reviewer #2: Authors are commended on their careful consideration and attention to revisions. No more revision suggestions.

7. PLOS authors have the option to publish the peer review history of their article (what does this mean?). If published, this will include your full peer review and any attached files.

Reviewer #2: No

---

## [Author Response · Author response to Decision Letter 2]

23 Apr 2025

We added the FM diagnostic criteria and citation requested.

---

## [Editor Report · Decision Letter 2]

Pain in the brain: Psychological correlates of chronic pain and fibromyalgia

PONE-D-24-49610R2

Dear Dr. Guerrieri,

We’re pleased to inform you that your manuscript has been judged scientifically suitable for publication and will be formally accepted for publication once it meets all outstanding technical requirements.

Kind regards,

Wesam Gouda, MD,PhD

Academic Editor

PLOS ONE
---

## [Editor Report · Acceptance letter]

PONE-D-24-49610R2

PLOS ONE

Dear Dr. Guerrieri,

I'm pleased to inform you that your manuscript has been deemed suitable for publication in PLOS ONE. Congratulations! Your manuscript is now being handed over to our production team.

Kind regards,

on behalf of

Dr. Wesam Gouda

Academic Editor

PLOS ONE